# Hard-label Manifolds: Unexpected advantages of query efficiency for finding on-manifold adversarial examples

## Abstract

Designing deep networks robust to adversarial examples remains an open problem. Likewise, recent zeroth order hard-label attacks on image classification tasks have shown comparable performance to their first-order alternatives. It is well known that in this setting, the adversary must search for the nearest decision boundary in a query-efficient manner. State-of-the-art (SotA) attacks rely on the concept of pixel grouping, or super-pixels, to perform efficient boundary search. It was recently shown in the first-order setting, that regular adversarial examples leave the data manifold, and on-manifold examples are generalization errors (Stutz et al., 2019). In this paper, we argue that query efficiency in the zeroth-order setting is connected to the adversary's traversal through the data manifold. In particular, query-efficient hard-label attacks have the unexpected advantage of finding adversarial examples close to the data manifold. We empirically demonstrate that against both natural and robustly trained models, an efficient zeroth-order attack produces samples with a progressively smaller manifold distance measure. Further, when a normal zeroth-order attack is made query-efficient through the use of pixel grouping, it can make up to a two-fold increase in query efficiency, and in some cases, reduce a sample's distance to the manifold by an order of magnitude.

## 1 Introduction

Adversarial examples in the context of deep learning models were originally investigated as blind spots in classification (Szegedy et al., 2013; Goodfellow et al., 2014). Formalized methods for discovering these blind spots emerged, referred to as gradient-level attacks, and became the first style to reach widespread attention by the deep learning community (Papernot et al., 2016; Moosavi-Dezfooli et al., 2015; Carlini & Wagner, 2016; 2017). In order to compute the necessary gradient information, such techniques required access to model parameters and a sizeable query budget, needing surrogate information to be competitive (Papernot et al., 2017). This naturally led to the creation of score-level attacks, which only require the confidence values output by machine learning models (Fredrikson et al., 2015; Tramèr et al., 2016; Chen et al., 2017; Ilyas et al., 2018). However, the grand prize of adversarial ML (AML), hard-label attacks, have been proposed in very recent years. These methods, which originated from a random-walk on the decision boundary (Brendel et al., 2017), have been carefully refined to offer convergence guarantees (Cheng et al., 2019), query efficiency (Chen et al., 2019; Cheng et al., 2020), and simplicity through super-pixel grouping (Chen & Gu, 2020), without ever sacrificing earlier advances.

Despite the steady improvements of hard-label attacks, open questions persist about their behavior, and AML attacks at large. The existence of adversarial samples were originally assumed to lie in rare pockets of the input space (Goodfellow et al., 2014), but this was later challenged by the boundary tilting assumption (Tanay & Griffin, 2016; Gilmer et al., 2018), which adopts a "data-geometric" view of the input space living on a manifold. This is supported by Stutz et al. (2019), who suggest that regular adversarial examples leave the data manifold, while on-manifold adversarial examples are generalization errors.

In this paper, we adopt the boundary-tilting assumption and demonstrate an unexpected benefit of query-efficient zeroth order attacks; such attacks are more likely to discover on-manifold examples,

and it is primarily enabled by the use of down-scaling techniques, such as super-pixel grouping. This is initially counter-intuitive, since down-scaling techniques reduce the search dimension, which artificially limits the search space, and can lead to worse (farther-away) adversarial examples. Our results suggest, however, that super-pixels help eliminate the search space of off-manifold adversarial examples, leading to examples which are **truly generalization errors**. With this knowledge, it is possible to rethink the design of hard-label attacks, towards those which resemble attack (b) in Figure 1, rather than (a) or (c).

Our specific contributions are as follows:

- **Reveal new insights of manifold feedback during query-efficient zeroth-order search.** We describe an approach for extending dimension-reduction technqiues in the score-level setting (Tu et al., 2019) to hard-label attacks. Afterwards, we propose the use of FID score (Heusel et al., 2018) as an $L_p$-agnostic means for estimating distance to the sampled submanifold. This measure allows to empirically demonstrate the connection between query efficiency and manifold feedback from the model, beyond the known convergence rates tied to dimensionality (Nesterov & Spokoiny, 2017). We later tie this result to known behavior in the gradient-level setting (Engstrom et al., 2019), which shows that manifold information is leaked from models in the hard-label setting, regardless of their robustness.

- **Attack-agnostic method for super-pixel grouping.** We show that bilinear down-scaling of the input space act as a form of super-pixel grouping, which yields up to 140% and 210% query efficiency gain for previously-proposed HSJA (Chen et al., 2019) and Sign-OPT attack (Cheng et al., 2020), respectively. These results align with previously observed behavior in the score-level setting (Tu et al., 2019).

- **Introduction of manifold distance oracle.** Our analysis covers a comprehensive array of datasets, model regularization methods, and $L_p$-norm settings from the literature. Regardless of the setting, we observe a consistent behavior of leveraging manifold information during query-efficient attacks. Thus we propose an information-theoretical formulation and interpretation of the *noisy manifold distance oracle*, which enables zeroth-order attacks to craft on-manifold examples. Studying this problem may assist in understanding the fundamental limits and utility of hard-label attacks.

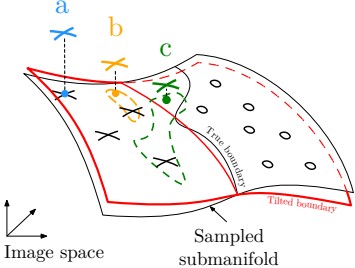

a: off-manifold, low-similarity
b: on-manifold, high-similarity
c: on-manifold, low-similarity
-·-·- -·-·-· : search space

Figure 1: Our interpretation of zeroth-order attack behavior in the context of boundary tilting (Tanay & Griffin, 2016): a) zeroth-order attack targeting low-level features, leaving the manifold, b) an efficient zeroth-order attack targeting mostly high-level features, floating along the manifold, and c) manifold-based zeroth-order attack next to the manifold, but sacrificing similarity.

## 2 RELATED WORK

Since the original discovery of adversarial samples (Szegedy et al., 2013; Goodfellow et al., 2014) and later formulations based on optimization (Carlini & Wagner, 2016; Moosavi-Dezfooli et al., 2015), the prevailing question was why such examples existed. The original assumption was that adversarial examples lived in low-probability pockets of the input space, and were thus never encountered during parameter optimization (Szegedy et al., 2013). This effect was believed to be amplified by the linearity of weight activations in the presence of small perturbations (Goodfellow et al., 2014). These assumptions were later challenged by the manifold assumption, which in summary claims that 1) the train and test sets of a model only occupy a sub-manifold of the true data, while the decision boundary lies close to samples on and beyond the sub-manifold (Tanay & Griffin, 2016), and 2) the "data geometric" view, where high-dimensional geometry of the true data manifold enables a low-probability error set to exist (Gilmer et al., 2018). Likewise the manifold assumption describes adversarial samples as leaving the manifold, which has inspired many defenses based on projecting such samples back to the data manifold (Jalal et al., 2019; Samangouei et al., 2018a), and adaptive attacks for foiling these defenses (Carlini et al., 2019; Carlini & Wagner, 2017; Tramer et al., 2020).

We investigate the scenario where an adversary uses zeroth-order information to either estimate the gradient direction, or find the closest decision boundary, related to previous work in the gradient-level setting by Stutz et al. (2019). In our setting, the adversary uses the top-1 label feedback from the model to reach their goal. They can also use a low-dimensional approximation of the data manifold to encourage query-efficiency. However, to date it is not completely understood how this affects the traversal through the data manifold, particularly in the zeroth-order setting.

## 3 ZEROTH-ORDER SEARCH THROUGH THE MANIFOLD

Our primary motivation is to characterize recent zeroth-order attacks as they relate to ideas of manifold traversal (Chen et al., 2019; Cheng et al., 2020; Chen & Gu, 2020). In the most common problem setting, the adversary is interested in attacking a $K$-way multi-class classification model $f : \mathbb{R}^d \rightarrow \{1, \ldots, K\}$. Given an original example $\mathbf{x}_0$, the goal is to generate adversarial example $\mathbf{x}$ such that $\mathbf{x}$ is close to $\mathbf{x}_0$ and $f(\mathbf{x}) \neq f(\mathbf{x}_0)$, where closeness is often approximated by the $L_p$-norm of $\mathbf{x} - \mathbf{x}_0$. The value of this approximation is debated in the literature (Heusel et al., 2018; Tsipras et al., 2018; Engstrom et al., 2019). Likewise we turn to alternative methods shown later for measuring closeness. First we step through the formulation for contemporary hard-label attacks, then show how dimension-reduced score-level attacks are extended to the hard-label setting, which enables analysis of zeroth-order decision boundary search.

### 3.1 GRADIENT-LEVEL FORMULATION

For gradient-level attacks, the goal is satisfied by first assuming that $f(\mathbf{x}) = \text{argmax}_i(Z(\mathbf{x})_i)$, where $Z(\mathbf{x}) \in \mathbb{R}^K$ is the final (logit) layer output, and $Z(\mathbf{x})_i$ is the prediction score for the $i$-th class, the stated goal is satisfied by the optimization problem,

$$h(\mathbf{x}) := \underset{\mathbf{x}}{\text{argmin}} \left\{ ||\mathbf{x} - \mathbf{x}_0||_p + c\mathcal{L}(Z(\mathbf{x})) \right\}, \tag{1}$$

for the Euclidean $L_p$-norm $|| \cdot ||_p$, $\mathcal{L}(\cdot)$ is the loss function corresponding to the goal of the attack, and $c$ is a regularization parameter. A popular choice of loss function is the Carlini & Wagner (2016) loss function.

### 3.2 SCORE-LEVEL AND HARD-LABEL ATTACKS

In the gradient-level setting, we require the gradient $\nabla f(\cdot)$. However, in the score-level setting we are forced to estimate $\frac{\partial f(\mathbf{x})}{\partial \mathbf{x}}$ without access to $\nabla f(\cdot)$, only evaluations of $Z(\cdot)$. Tu et al. (2019) reformulate the previous problem to a version relying instead on the ranking of class predictions from $Z$. In practical scenarios, the estimate is found using random gradient-free method (RGF), a scaled random full gradient estimator of $\nabla f(\mathbf{x})$, over $q$ random directions $\{\mathbf{u}_i\}_{i=1}^q$. The score-level setting was extended to several renditions of the hard-label setting, which we clarify below. In each case the goal is to approximate the gradient by $\hat{\mathbf{g}}$.

**OPT-Attack** For given example $\mathbf{x}_0$, true label $y_0$, and hard-label black-box function $f : \mathbb{R}^d \rightarrow \{1, \ldots, K\}$, Cheng et al. (2019) define the objective function $g : \mathbb{R}^d \rightarrow \mathbb{R}$ as a function of search direction $\boldsymbol{\theta}$, where $g(\boldsymbol{\theta}^*)$ is the minimum distance from $\mathbf{x}_0$ to the nearest adversarial example along the direction $\boldsymbol{\theta}$. For the untargeted attack, $g(\boldsymbol{\theta})$ corresponds to the distance to the decision boundary along direction $\boldsymbol{\theta}$, and allows for estimating the gradient as,

$$\hat{\mathbf{g}} = \frac{1}{q} \sum_{i=0}^{q} \frac{g(\boldsymbol{\theta} + \beta\mathbf{u}_i) - g(\boldsymbol{\theta})}{\beta} \cdot \mathbf{u}_i. \tag{2}$$

where $\beta$ is a small smoothing parameter. Notably, $g(\boldsymbol{\theta})$ is continuous even if $f$ is a non-continuous step function.

**Sign-OPT** Cheng et al. (2020) later improved the query efficiency by only considering the sign of the gradient estimate,

$$\hat{\nabla} g(\boldsymbol{\theta}) \approx \hat{\mathbf{g}} := \sum_{i=1}^{q} \mathrm{sgn}\left(g(\boldsymbol{\theta} + \beta \mathbf{u}_i) - g(\boldsymbol{\theta})\right) \mathbf{u}_i.$$

We focus on the Sign-OPT variant, as the findings are more relevant to the current SotA.

**HopSkipJumpAttack** Similar to Sign-OPT, HopSkipJumpAttack (HSJA) (Chen et al., 2019) uses a zeroth-order sign oracle to improve Boundary Attack proposed by Brendel et al. (2017). HSJA lacks the convergence analysis of OPT Attack/Sign-OPT and relies on one-point gradient estimate. Regardless, HSJA is competitive with Sign-OPT for SotA in the $L_2$ setting.

**RayS** Chen & Gu (2020) propose an alternative method which is to search for the minimum decision boundary radius $r$ from a sample $\mathbf{x}_0$, along a ray direction $\boldsymbol{\theta}$. Instead of searching over $\mathbb{R}^d$ to minimize $g(\boldsymbol{\theta})$, Chen et al. propose to perform ray search over directions $\boldsymbol{\theta} \in \{-1, 1\}^d$, resulting in $2^d$ maximum possible directions. This reduction of the search resolution enables SotA query efficiency in the $L_\infty$ setting with proof of convergence. The search resolution is further reduced by the hierarchical variant of RayS, which performs on-the-fly upscaling of image super-pixels.

### 3.3 DIMENSION-REDUCED ZEROTH-ORDER SEARCH

The attacks described so far each represent an improvement in query efficiency under different $L_p$-norm scenarios. The difficulty in performing a holistic analysis of their behavior lies in each attack's unique design. In order to characterize query-efficient attacks in a consistent way, we rely on a hard-label version of the reduced-dimension scheme proposed by Tu et al. (2019). This scheme can allow to dynamically scale the expected query-efficiency up or down in a controlled manner. The reduced-dimension search is feasible since it was shown that the intrinsic dimensionality of data is often lower than the true dimension (Amsaleg et al., 2017). Likewise, zeroth-order attacks can exploit the known convergence rate of zeroth order gradient descent, which is tied to a dimensionality $d$ of the vectorized input (Nesterov & Spokoiny, 2017; Liu et al., 2018). In practice this reduction is implemented through an encoding map $\mathcal{E} : \mathbb{R}^d \to \mathbb{R}^{d'}$ for reduced dimension $d'$ and decoding map $\mathcal{D} : \mathbb{R}^{d'} \to \mathbb{R}^d$. In general the adversarial sample is created by

$$\mathbf{x} = \mathbf{x}_0 + g\left(\mathcal{D}(\boldsymbol{\theta}')\right) \frac{\mathcal{D}(\boldsymbol{\theta}')}{||\mathcal{D}(\boldsymbol{\theta}')||}, \tag{3}$$

where $\boldsymbol{\theta}' \in \mathbb{R}^{d'}$ and is optimized depending on the respective attack (e.g., Sign-OPT and HSJA), and as before, $g$ is a measure of distance to the decision boundary in direction $\mathcal{D}(\boldsymbol{\theta}')$. The mapping functions can be initialized with either an autoencoder (AE), or a pair of channel-wise bilinear transform functions (henceforth referred to as BiLN) which simply scales the input up or down depending on a fixed scaling factor. These choices were previously investigated by Tu et al. (2019) as a way to improve query efficiency in score-level attacks, and ultimately performed similarly with respect to query efficiency. For our purposes, these functions represent two distinct methods of synthesizing adversarial samples, which either rely on an approximate description of the manifold (AE), or instead use a deterministic rescaling to achieve efficiency (BiLN). The inclusion of BiLN is important, because it allows to measure the scenario where the adversary has no explicit knowledge of the manifold, and only relies on the feedback from the model.

Under the AE scenario, the AE is tuned to minimize reconstruction error of input images. Due to this dependence on labeled data, the output quality of the AE is dependent on the adversary's ability to collect data, which is a realistic consideration. We model the scenario where the adversary only has access to the test set, which is often considerably less informative than the training set. This manifests as an extra distortion in addition to the adversarial noise. Thus the output of the AE-initialized decoder can be used in different ways, which we discuss briefly in Section A.3 of the Appendix. In the case of BiLN, no additional training is required, which means it synthesizes search direction independent of the adversary's manifold description (i.e., possible extracted knowledge about test samples). This can manifest as a lower overall distortion, in the case where only a crude manifold

description can be extracted from the test set. Next we describe the exact usage of the mapping functions for each attack scheme.

**Sign-OPT & HSJA.** In general for the attacks which rely on the Cheng et al. (2019) formulation, the update in Equation 2 becomes

$$\hat{\mathbf{g}} = \frac{1}{q} \sum_{i=0}^{q} \frac{g(\boldsymbol{\theta}' + \beta \mathbf{u}_i') - g(\boldsymbol{\theta}')}{\beta} \cdot \mathbf{u}_i', \tag{4}$$

for the reduced-dimension Gaussian vectors $\{\mathbf{u}_i' \in \mathbb{R}^{d'}\}_{i=0}^{q}$ and direction $\boldsymbol{\theta}' \in \mathbb{R}^{d'}$ for integer $d' < d$. The reduced-dimension direction $\boldsymbol{\theta}'$ is initialized randomly with $\boldsymbol{\theta}' \sim \mathcal{N}(0, 1)$ for the untargeted case, or for the targeted case as $\boldsymbol{\theta}' = \mathcal{E}(\mathbf{x}_t)$, where $\mathbf{x}_t$ is a test sample correctly classified as target class $t$ by the victim model. This scheme also applies to HSJA, since HSJA performs a single-point sign estimate. As in the normal variants, $\hat{\mathbf{g}}$ is used to update $\boldsymbol{\theta}'$.

**RayS.** The intuition behind RayS attack is to perform a discrete search in at most $2^d$ directions. Chen et al. also perform a hierarchical search over progressively larger super-pixels of the image. This has the effect of already upscaling *on-the-fly* (Chen & Gu, 2020). The inclusion of RayS in our analysis is important, since it has the unique behavior of performing a discrete search for the decision boundary, rather than an explicit gradient estimate. To achieve an appropriate reduced-dimension version of RayS, and test our hypothesis, we modify the calculation of $s$ in Algorithm 3 of Chen & Gu (2020), which either speeds up upscaling by a factor $a$ (i.e., $s = s + a$), or extends the search through a specific block index by a factor $b$ (increase block level at $k = 2^s b$ instead of $k = 2^s$).

### 3.4 Capturing zeroth-order search deviation

Our analysis follows in the wake of findings presented by (Stutz et al., 2019). It is shown that "regular" gradient-level adversarial examples leave the data manifold i.e., the sample's distance to the manifold is larger than with an "on-manifold" gradient-level adversarial example. In the score-level and hard-label settings, the manifold can be used to guide the search for the boundary. The benefits of this scenario can be observed in score-level attacks by Tu et al. (2019). Similarly, hard-label attacks can leverage the concept of super-pixels to achieve gains in performance (Chen & Gu, 2020). However, to date the connection between dimension manipulation and manifold traversal has not been investigated. The reduced search fidelity naturally limits the resolution of search direction, which is the final noise vector applied to the original sample, as evidenced by Equation 3. This introduces our central research question: *How does searching over reduced-dimension increase efficiency, if the search resolution is decreased as a side-effect?*

To approach this question, we observe that certain gradient-level and score-level attacks can leverage a prior manifold description to attack more efficiently (Stutz et al., 2019; Tu et al., 2019). We can rely on a sample's distance to the manifold as a measure of deviation during the search, similar to the work by Stutz et al. (2019). Hereafter, we refer to this distance *w.l.o.g* as the manifold distance. Our choice is motivated by the fact that from a data-geometric perspective, manifold distance describes the amount of semantic features preserved during the attack process (Engstrom et al., 2019). Likewise the manifold distance can communicate information about attack behavior better than $L_p$ distortion measurements, which are the common choice in existing zeroth-order attack literature (Cheng et al., 2020; Chen et al., 2019; Chen & Gu, 2020). Second, the manifold approximation technique by Stutz et al. (2019) is mainly suited for $L_2$-norm, whereas hard-label attacks exist for both $L_2$ and $L_\infty$-norm. Unfortunately, the real data manifold is difficult to describe. This is an open problem in the study of Generative Adversarial Networks (GANs), as designers must ensure that generator images are on-manifold using a continuous function. This has motivated the recently proposed Frechet Inception Distance (FID), which acts as a surrogate measure of the manifold distance (Heusel et al., 2018). We can leverage FID by treating the adversarial samples as synthetically generated images. FID is viable as it computes Frechet distance of candidate images with respect to images sampled from the true data sub-manifold. Since FID uses an Inception-V3 coding layer to encode images, this distance will correlate with distortion of high-level features, which will be a result of sampling farther from the data manifold. We do not target the Inception-V3 network in any of our experiments, so the FID metric will not rely on any internal aspects of the victim models.

## 4 RESULTS

### 4.1 METHODOLOGY

Our experimental analysis addresses the following three research questions about zeroth-order attacks:

Q1. What is the trade-off between query efficiency and reduced search resolution in the zeroth order setting, against both natural and "robust" models?

Q2. In addition to reducing queries, are there unexpected benefits for performing a query-efficient zeroth order attack?

Q3. Compared to the previous results in the score-level setting (Tu et al., 2019), do dimension-reduced hard-label attacks produce a similar amount of reduction in query usage?

We answer these questions by comparing three hard-label attacks against their dimension-reduced variants. Some variants are not shown due to incompatibility with the base attack. For example, AE+HSJA is not implemented as it relies on only a single-point estimate, thereby only allowing to attack on the manifold directly. This is not practical due to induced distortion discussed previously in Section 3. RayS can perform two-point search, but it assumes codependence of input features, which may not be the case for well-defined latent space of an autoencoder.[1] Thus for the AE variant, we rely on Sign-OPT as it can perform two-point estimate and does not rely on codependence of features. In the BiLN cases, the implementations follow the discussion in Section 3.3.

**Experimental Highlights.** Our experiments show that query-efficient attacks exhibit unexpected behaviors and benefits, with explanations summarized below:

A1. Reduced search resolution increases search efficiency by allowing the discovery of samples not encountered during training. This is especially true for robust models, which we show have a tendency to overfit on the first-order noise characteristics.

A2. Surprisingly, query-efficient attacks search closer to the manifold than non-efficient attacks, thus are more likely to produce on-manifold examples. This occurs because query-efficient search acts as indirect manifold feedback. By reducing the search fidelity, we are more likely to modify high-level features. This effect is magnified on robust models, which are already known to leak manifold information in gradient-level settings (Engstrom et al., 2019).

A3. Dimension-reduced attacks are capable of SotA query-efficiency gains for HSJA, and a two-fold improvement for Sign-OPT.

**Setup.** All attacks run for 25k queries without early stopping. For brevity, we only show results for the untargeted case. FID score is calculated using the 64-dimensional max pooling layer of the Inception-V3 deep network for coding (denoted as FID-64 in figures), taken from an open-source implementation.[2] The choice of the 64-dimensional feature layer allows to calculate full-rank FID without the full 2,048 sample count of original FID, which is prohibitive given the high time complexity of chosen robust models for analysis (in particular, certified smoothing ImageNet). This incurs the cost of losing some high-level features, but due to the position of the chosen coding layer in the network, it is still valuable for direct comparison of manifold distance between attacks. Since the coding layer differs from the original implementation, the magnitudes will differ from those published by Heusel et al. (2018).

Image data consists of the CIFAR-10 (Krizhevsky, 2009) and ImageNet (Russakovsky et al., 2015) classification datasets. This selection allows to study attack behavior on both small and high-resolution image data. Original samples are chosen from the test set of each dataset using a similar technique as in Chen et al. (2019). On CIFAR-10, ten random samples are taken from each of ten classes. On ImageNet, ten random classes are chosen with ten random samples taken from each (i.e., 100 total samples on either dataset). We provide further implementation details in Section A.3 of the Appendix. In addition to natural images, we are interested in the attack behavior for a model regularized with some variant of first-order noise. For CIFAR-10, we choose the representative adversarial training

---

[1] Experimentally, codependence hurt the AE+RayS variant more than was practically useful.

[2] https://github.com/mseitzer/pytorch-fid

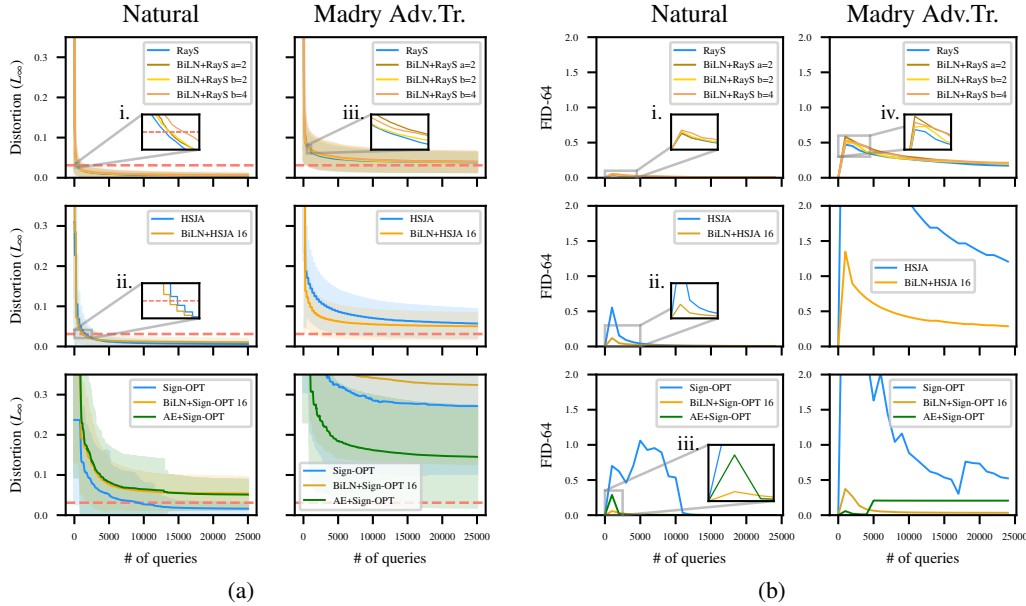

Figure 2: Results across attacks for CIFAR-10 dataset, corresponding to a) distortion against query usage (dotted red line denotes the value of $\epsilon$, shaded areas mark standard deviation), and b) FID-64 trajectory against the same query usage.

technique proposed by Madry et al. (2017). For completeness, we also include ablation results on more recent regularization techniques in Section A.5 of the Appendix. For ImageNet, we compare against the SotA at time of writing, randomized smoothing proposed by Cohen et al. (2019). We use the pre-trained Resnet50 weights and implementation provided by Cohen et al., corresponding to smoothing parameter $\sigma = 0.5$ and $\epsilon \simeq 1.0$.

## 4.2 EXPERIMENTAL DETAILS

On each dataset, we target the $L_p$-norm which the robust models were regularized or certified under. Likewise we use the $L_\infty$ versions of each attack for CIFAR-10 and $L_2$ versions for ImageNet.

**CIFAR-10 case study ($L_\infty$).** We start by measuring the distortion against remaining query budget of the adversary in Figure 2a. In general, the normal variants of each attack align with the published results. The BiLN variants of RayS each have minimal effect on overall query efficiency (Insets 2a.i and 2a.iii). This is a result of RayS not relying on explicit gradient estimation. The main improvement is with BiLN+HSJA against the Madry adversarial training model, with average distortion at 4k queries decreasing from 0.09 to 0.07, and corresponding success rate increasing from 15% to 25%. This improvement aligns with the result of Tu et al. (2019), and is contrary to the minimal effect on the natural model (Inset 2a.ii). AE+Sign-OPT outperforms against regular Sign-OPT and BiLN+Sign-OPT on the robust model, since searching along the manifold can grant distortion which was not encountered during first-order regularization (Stutz et al., 2019; Jalal et al., 2019). However, the success with AE+Sign-OPT tends to be situational, whereas HSJA and RayS outperform in either scenario. Mainly the low quality of the adversary's AE does not permit fine-grained adjustment in latent space that RayS and HSJA can provide in image space. We next focus on Figure 2b, which shows the FID score's trajectory as the search progresses. Every trajectory will begin at a zero value, since there is an expected score of zero for identical images, and then peaks as the attack initialization is performed. In this scenario, our main observation is the similarity of FID trajectory between RayS, HSJA, and dimension-reduced Sign-OPT, despite the method of model regularization, as shown in Insets 2b.i, 2b.ii, and 2b.iii, respectively. For example, the magnitudes for AE+Sign-OPT and RayS peak to 0.28 and 0.04, then fall (and stay) at values near 0.004. BiLN+HSJA and BiLN+Sign-OPT both exhibit lower FID scores than their normal variants, as much as two orders of magnitude less in

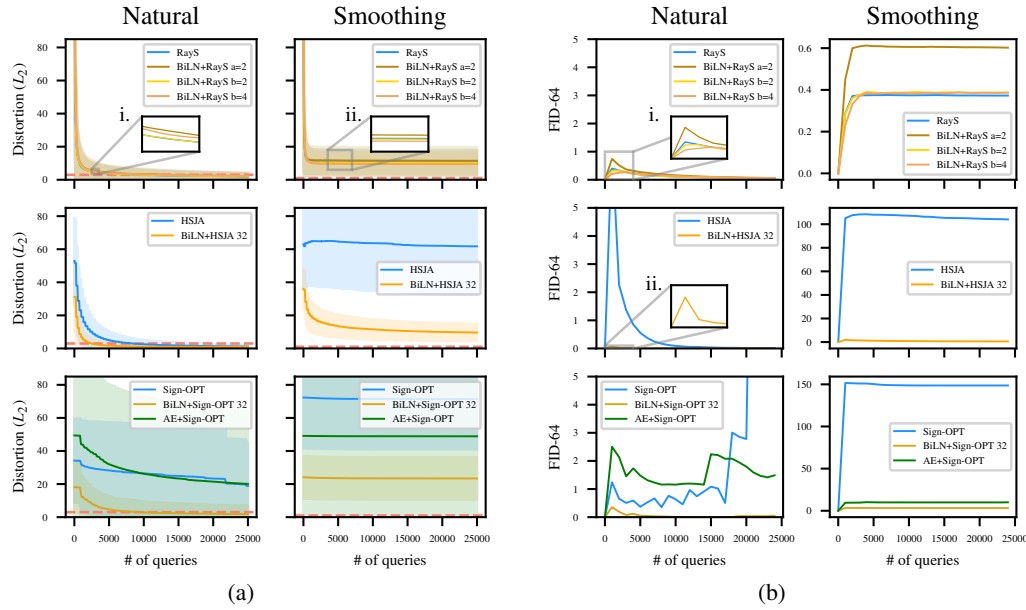

Figure 3: Results across attacks for ImageNet dataset, corresponding to a) distortion against query usage (dotted red line denotes the value of $\epsilon$, shaded areas mark standard deviation), and b) FID-64 trajectory against the same query usage.

the case of Sign-OPT and BiLN+Sign-OPT. For a detailed comparison of distortion and FID values for CIFAR-10, we offer Table 1 in Section A.5 of the Appendix. The dimension-reduced variants of RayS do not have a large variation between them (Inset 2b.iv), a side-effect of the adaptive super-pixel search, which can automatically scale the super-pixel size as the search progresses. At a high level, we observe that BiLN+HSJA offers comparable performance to regular RayS, while in the robust case, BiLN and AE variants tend towards lower FID scores than regular variants. This can be viewed as the model leaking manifold information through the decision, as was shown for the first-order gradient by Engstrom et al. (2019).

**ImageNet case study** ($L_2$). We attack ImageNet in the $L_2$-norm setting to compare against the certified smoothing technique proposed by Cohen et al. (2019). The label output comes from a smooth classifier, approximated by many rounds of Monte Carlo search, which uses the regular model regularized by Gaussian noise. Thus hard-label attacks are well positioned for attacking the smoothing technique. The distortion results of these attacks are shown in Figure 3a. Dimension reduction has a larger impact when coupled with the large ImageNet resolution. Particularly the BiLN+HSJA and BiLN+Sign-OPT attacks profit the most. At 8k queries, success rate increases 1.4x and 2.1x for HSJA and Sign-OPT, respectively. As before with CIFAR-10, the RayS dimension reduction is saturated on ImageNet (Insets 3a.i and 3a.ii). Since RayS does not explicitly rely on gradient estimation, it benefits the least from dimension-reduction techniques. Apart from RayS, BiLN variants outperform both AE and regular variants in every scenario, despite the reduced search fidelity. The improvement is largest on the smoothing technique, particularly with HSJA and Sign-OPT (second column of Figure 3b). This is due to 1) the adversary's reduced description of the manifold, by only having access to the test set, and 2) BiLN allowing to search closer to the original sample, since it is a deterministic function independent of the adversary's knowledge. The FID scores in Figure 3b paint a more comprehensive picture. BiLN variants produce adversarial examples closer to the manifold than either regular or AE variants, highlighted with HSJA+BiLN in Inset 3b.ii. RayS saw no such improvement, as it does not perform an explicit gradient estimate (Inset 3b.i). We interpret this as follows: BiLN variants on HSJA and Sign-OPT leverage reduced fidelity to increase the probability of finding the decision boundary, and 1) produce a smoother noise distribution, resulting in more spatially correlated distortion, which as a result 2) produces adversarial examples closer to the manifold. Since the manifold description of ImageNet is crude, BiLN variants can excel, since they search independent of

this description. Another key observation is the fluctuation of LID score towards the end of Sign-OPT and AE+Sign-OPT, which are not present for HSJA or RayS (first column of Figure 3b). Notably there is no direct signal of manifold distance for the normal or BiLN variants. This indicates that query-efficient attacks do a better job of capturing implicit manifold distance feedback from the model.

## 5    DISCUSSION

**The noisy manifold distance oracle.**    In Section 4 we observed that query-efficient attacks, i.e., those which leverage the concept of super-pixels to reduce search fidelity, are more likely to produce samples close to the manifold. This generates samples that are unseen by robust models during their first-order adversarial training. However, this takes place without any direct feedback about manifold distance. To approach this, we first consider that the model relies on a subsampled manifold of the image space. This manifold can be leaked by the loss landscape of the model, as shown by Engstrom et al. (2019). From an information-theoretical perspective, the zeroth-order adversary observes the *noisy gradient*, which is leaked as side information by each model decision. Under this explanation, the decision feedback by the model is viewed as a noisy manifold distance (NMD) oracle. The improvement of AE+Sign-OPT on robust CIFAR-10 can be argued as a result of the NMD oracle improving as well. This can be shown using the data processing inequality (Beaudry & Renner, 2012): if $I(\mathcal{M}, \mathbf{g})$ increases, then $I(\mathcal{M}, \ddot{\mathbf{g}})$ also increases, where I is mutual information, $\mathcal{M}$ is the manifold, and $\ddot{\mathbf{g}}$ is the noisy gradient. In words, the quality of the noisy gradient depends on the quality of the model's loss landscape, which can more closely resemble the manifold under robust regularization. This means a higher quality loss landscape leads to a higher quality zeroth-order attack. Qualitative evidence of this effect can be observed in Section A.4 of the Appendix.

**"Topology" of hard-label settings.**    We can view zeroth-order attacks as following a topological hierarchy that is a function of the original data dimension. A very simple version of this idea is illustrated in Figure 1. Each technique illustrated in Figure 1 offers a different traversal distance both along the manifold, and away from it. Efficient attacks represented by (b) can combine elements of staying near the manifold, and traversing it. This is best seen by the results of BiLN variants and regular RayS. RayS succeeds because it assumes spatial correlation, which can be considered a loose description of the manifold, without making further assumptions of the data. Thus we can expect this behavior from future attacks which do not perform explicit gradient estimation. Similarly, purely traversing close to an ideal manifold description, as in (c), may not be advantageous, because a crude manifold approximation induces its own error. Further, the nearest boundary on the manifold could be far away. Thus we can consider an attack which leverages an ideal "manifold description" (e.g., through an autoencoder), but leverages the description as a method for selecting progressively smaller super-pixel groupings. For instance, RayS selects super-pixels as blocks, which are bifurcated until the number of super-pixels equals the original dimension. Notably, RayS displayed little to no benefit from dimension-reduction, which is a byproduct of not relying on explicit gradient estimation. A technique which leverages the manifold description to select super-pixel groups could yield better performance, both in terms of distortion and FID score. To this end, we feel the FID score offers a valuable measure of manifold distance, which can inform the topological behavior, and the quality of future hard-label attacks.

## 6    CONCLUSION

Despite the recent progress in zeroth-order attack methods, open questions remain about their precise behavior. We shed light on an unexpected nuance, which is their ability to produce on-manifold adversarial examples. This is despite the absence of manifold distance information, which motivates the proposal of the noisy manifold distance oracle. Future work could create a formal definition of this oracle, and attempt to bound the information revealed by the oracle. On the other hand, with knowledge of the oracle's existence, it is possible to further refine hard-label attacks, so they continue to reveal insights into the weaknesses of learning systems.

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

# A APPENDIX

## A.1 MNIST ABLATION STUDY

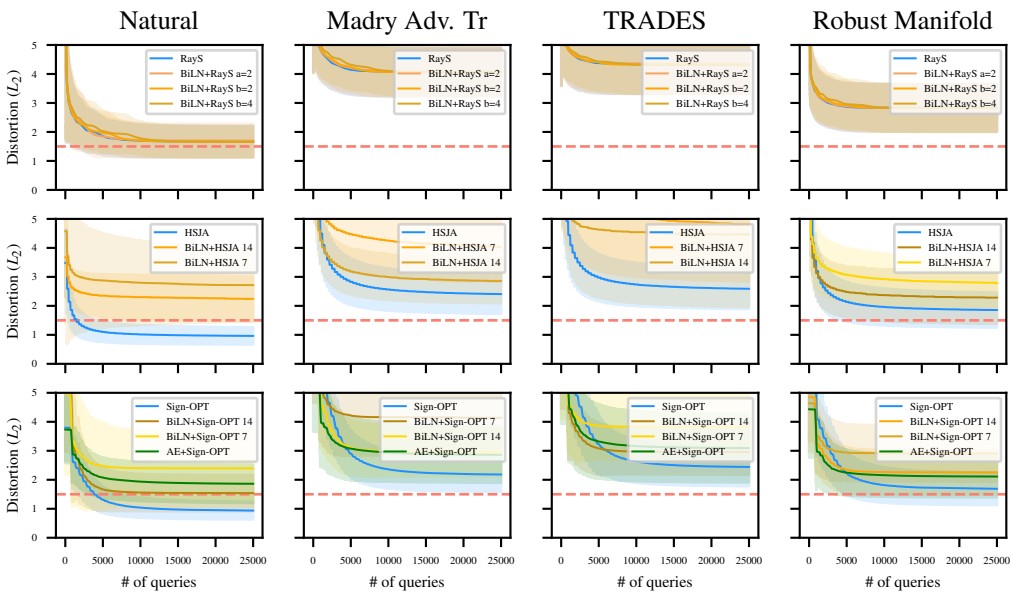

Figure 4: Results of image space distortion for the MNIST ablation on the natural model, manifold projection defense (Robust Manifold) and non-projection baselines (Madry adversarial training and TRADES). Dotted red line denotes the value of $\epsilon$, shaded areas mark standard deviation.

Manifold projection was originally proposed by Samangouei et al. (2018b) as a scheme to defend models in the gradient-level setting, and was partially broken by Athalye et al. (2018) due to the imperfect projection of the defender decoder. Jalal et al. (2019) later showed it could be circumvented completely, by searching for latent sample pairs which are close to each other in latent space, but far apart on the model loss landscape (Jalal et al., 2019). These sample pairs are projected to input space and used for adversarial training, which yields the Robust Manifold defense. In direct comparisons under the first-order setting, Jalal et al. (2019) show that the Robust Manifold scheme defends better than previous baselines, Madry adversarial training (Madry et al., 2017) and TRADES (Zhang & Wang, 2019), which do not leverage the manifold projection. We conduct experiments to directly compare this behavior in the hard-label setting, shown in Figure 4. The hard-label attacks consistently outperform against the Robust Manifold defense, despite this defense outperforming both the Madry and TRADES defenses under gradient-level attacks. This can be considered a result of the Robust Manifold defense overfitting to the manifold projection of the fixed generator (or "spanner" in the vocabulary of Jalal et al.). Further, dimension-reduced variants yield better exploration, shown by consistently better performance from 0-5000 queries. This supports the results of the main paper, which show that these defenses can overfit to first-order information.

The results on the MNIST ablation also support the notion of a dynamic topological hierarchy, which is dependent on the original data dimension $d$. In the case of HSJA (second row of Figure 4), query-efficient attacks do not benefit the adversary, because MNIST is already sufficiently low-dimension to yield samples near the manifold with regular hard-label attacks. We can validate this finding by comparing a notion of manifold distance between the attacks. FID is not suitable for MNIST as it requires re-training of the Inception-V3 network and calibration for use as a metric, which is outside the scope of this work. Instead we project the adversarial samples back to the manifold, using the Robust Manifold defender's variational autoencoder (VAE), and measure the pairwise distances between original and adversarial projections. This distance is described formally as $||\mathcal{D}(\mathcal{E}(\mathbf{x}_0)) - \mathcal{D}(\mathcal{E}(\mathbf{x}))||_2$, for each adversarial sample $\mathbf{x}$ and original sample $\mathbf{x}_0$. We use the checkpoint provided by Jalal et al. (2019). Notably the defender network has the best description of

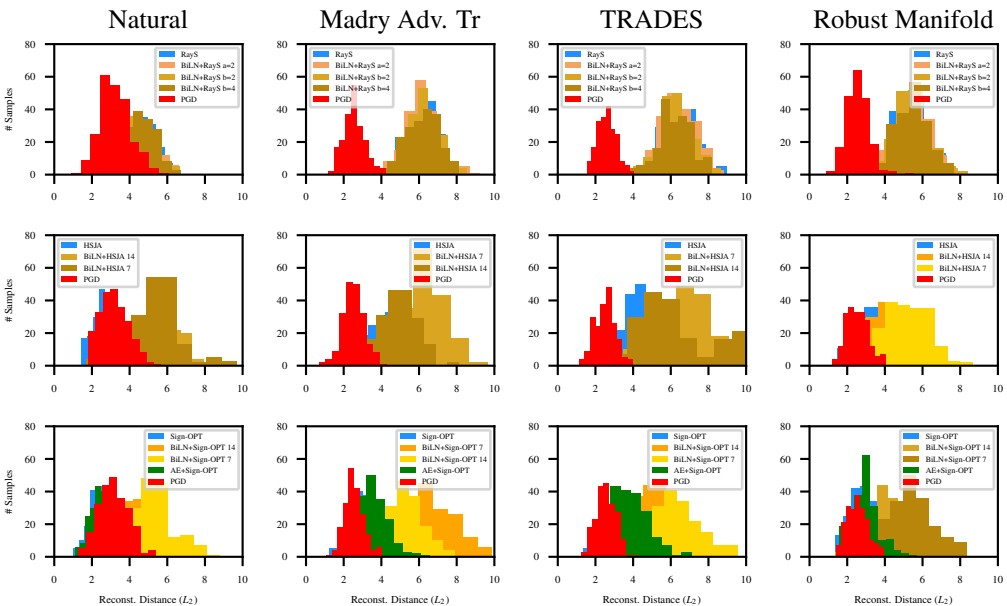

Figure 5: Measurement of pairwise reconstruction distance $||\mathcal{D}(\mathcal{E}(\mathbf{x}_0)) - \mathcal{D}(\mathcal{E}(\mathbf{x}))||_2$ for the MNIST ablation on the natural model, manifold projection defense (Robust Manifold) and non-projection baselines (Madry adversarial training and TRADES), using the original VAE checkpoint provided by Jalal et al. (2019) for $\mathcal{E}$ and $\mathcal{D}$.

the manifold, since it has access to the full training set. The distribution of pair-wise distances for each attack and defense combination is shown in Figure 5. In these plots, we expect the reconstruction distance to be smaller for adversarial samples closer to the manifold, since the VAE will capture high-level semantic information of the dataset.

Under our hypothesis, successful hard-label attacks should have a smaller distance on natural models than regular white-box attacks. Thus we also compare against samples generated with Projected Gradient Descent (PGD) in red. The results in Figure 5 support this hypothesis, as both Sign-OPT and HSJA have smaller distance than the PGD samples. This is exaggerated with the AE+Sign-OPT variant, due to the manifold approximation of the adversary. On robust models (second to last columns), PGD acts as a best-case for the adversary, since it is known from previous results by Engstrom et al. (2019) that gradient-level attacks on robust models leak manifold information. As discussed in the main text, hard-label attacks receive a noisy version of this manifold feedback, which is sufficient to create samples near the manifold. This is supported by the pairwise distances on robust models, since PGD samples are consistently closer to regular samples. Due to the dynamics of the topological hierarchy discussed before, the dimension-reduction is less effective on MNIST. Notably the regular hard-label attack variants (shaded blue) can be closer to the original samples than their dimension-reduced variants. RayS (top row) is consistently farther from the PGD samples, as it does not explicitly rely on gradient estimation like Sign-OPT and HSJA. We further point out that despite the variability of distortion in image space between regular and BiLN variants in Figure 4, these samples still lie near each other when projected to the manifold in Figure 5. On the Robust Manifold defense, HSJA and Sign-OPT attacks, regardless of variant, will lie closer to the PGD samples. This supports our main result, that query-efficient hard-label attacks leverage manifold information to produce samples, particularly against robust models.

## A.2    SUCCESS RATE PLOTS

In Figure 6 we provide query vs. success rate to accompany the results in the main text.

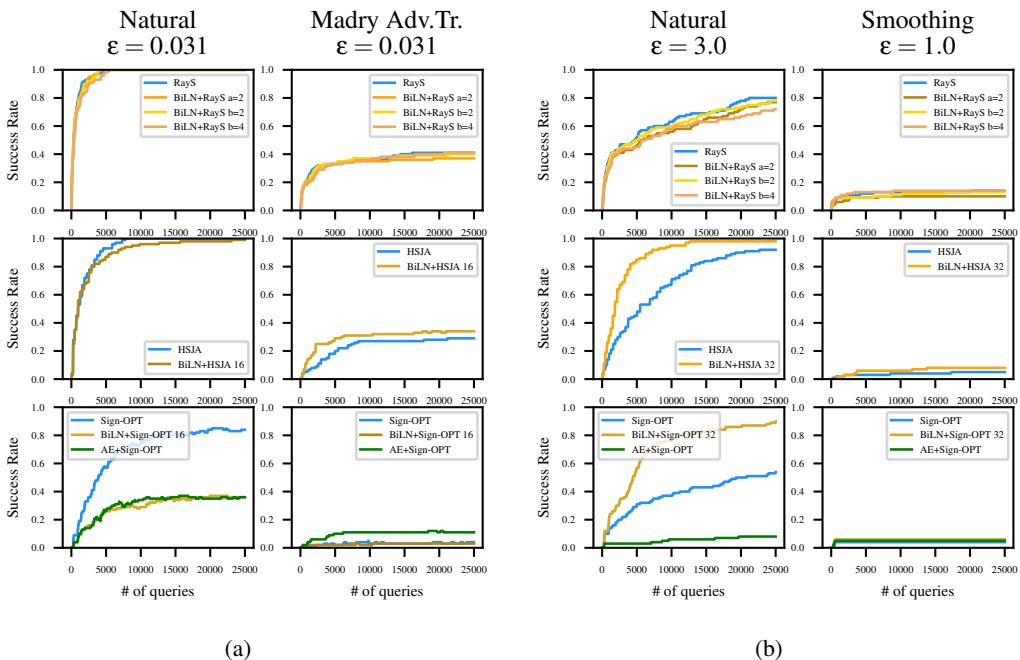

Figure 6: Query vs. success rate plots corresponding to each attack variant in the main text, for a) CIFAR-10 and b) ImageNet.

## A.3 IMPLEMENTATION DETAILS

We are primarily interested in the effect of reduced search resolution on attack behavior. Thus in this work, given a candidate direction $\boldsymbol{\theta}'$ and magnitude (or radius) $r$, the adversarial sample in the AE case is the blending $(1 - r)\mathbf{x}_0 + r\mathcal{D}\left(\mathcal{E}(\mathbf{x}_0) + \boldsymbol{\theta}'\right)$.[3]

Natural victim architectures consist of a deep convolutional neural network for CIFAR-10, and a Resnet50 network for ImageNet. The CIFAR-10 network is the same implementation open-sourced by Cheng et al. (2020), while the Resnet50 network is taken from the PyTorch Torchvision library, including pre-trained weights.[4] For AE attack variants, we implement the same architecture described by Tu et al. (2019). Specifically it leverages a fully convolutional network for the encoder and decoder. ImageNet samples are downsized to 128x128 before passing to the encoder, and the output of the decoder is scaled back to 224x224, as described by Tu et al. (2019). Every AE is trained using the held out test set, as we assume disjoint data between adversary and victim. To avoid ambiguity, we label each BiLN variant with the spatial dimension after performing the bilinear transformation, and keep this variable fixed for simplicity.

## A.4 VISUAL RESULTS - CIFAR10

We provide visual qualitative results for each attack on CIFAR-10 in Figure 7.

---

[3]We observed that it is detrimental to set $\mathbf{x} = \mathcal{D}(\mathcal{E}(\mathbf{x}_0) + r\boldsymbol{\theta}')$ or $\mathbf{x} = \mathcal{D}(r\boldsymbol{\theta}')$ directly. Despite remaining on the data manifold by attacking it directly, the approximation of the data manifold is crude, which results in large distortion (Stutz et al., 2019).

[4]https://pytorch.org/docs/stable/torchvision/models.html

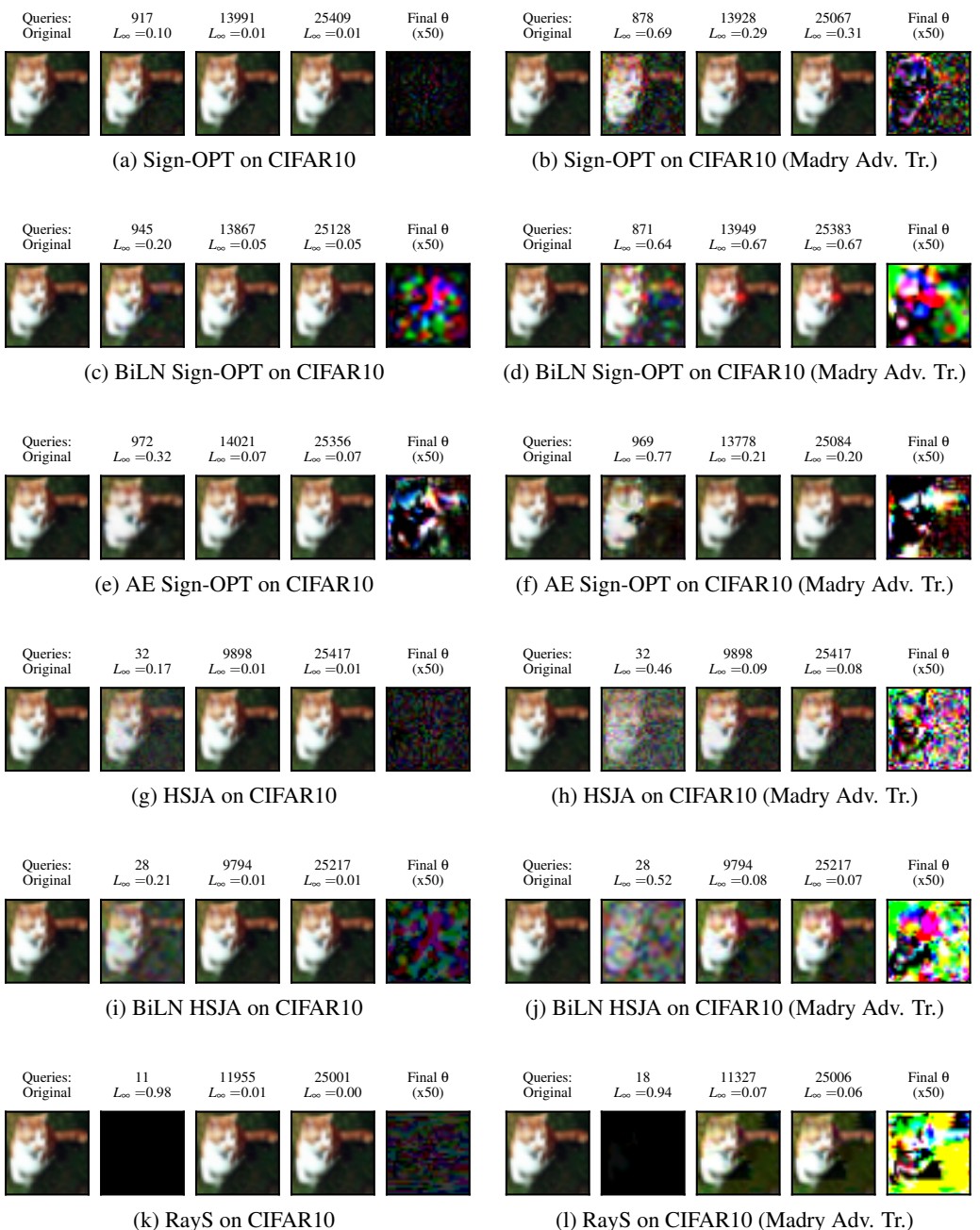

Figure 7: Visual selection of attack trajectories on CIFAR-10.

## A.5 SUPPLEMENTAL RESULTS

### A.5.1 EXTRA EXPERIMENTS ON ROBUST CIFAR-10

In Figures 8 and 9, we provide results on four extra model regularization methods from the literature: TRADES (Zhang et al., 2019), adversarial interpolation learning (Zhang & Xu, 2020), feature scattering (Zhang & Wang, 2019), and SENSE (Jungeum & Wang, 2020).

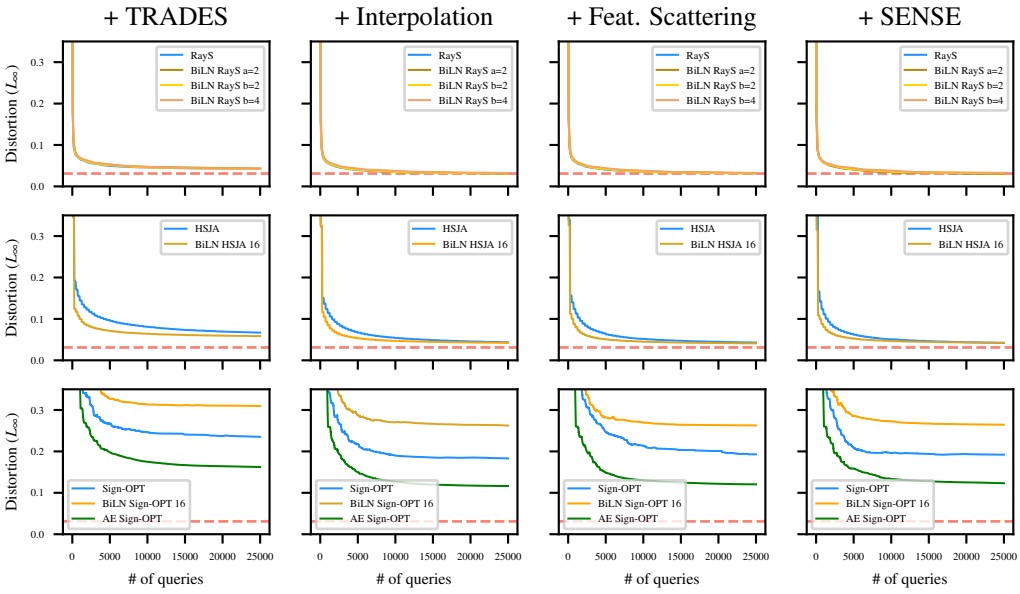

Figure 8: Results of distortion for CIFAR10 with four other choices of robust regularization from the literature.

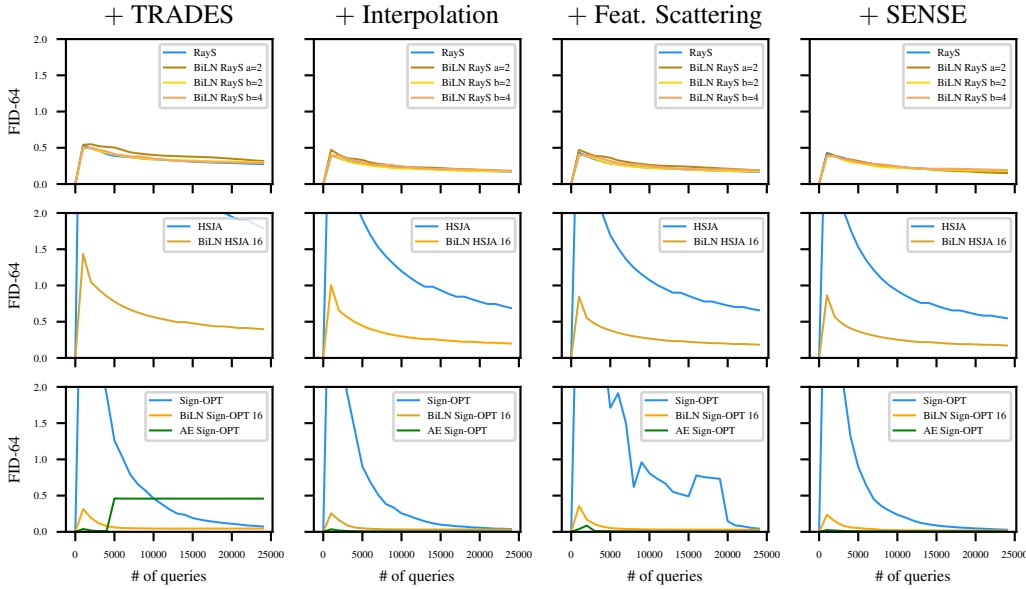

Figure 9: Results of FID score for CIFAR10 with four other choices of robust regularization from the literature.

| | CIFAR-10 ($\epsilon = 0.031$) | | | | + Madry Adv.Tr. ($\epsilon = 0.031$) | | | |
|---|---|---|---|---|---|---|---|---|
| | # Queries | Avg. $L_\infty$ | SR | FID-64 | # Queries | Avg. $L_\infty$ | SR | FID-64 |
| RayS | 4,000 | 0.01 | 98.0 | 0.02 | 4,000 | 0.05 | 33.0 | 0.36 |
| | 8,000 | 0.01 | 100.0 | 0.01 | 8,000 | 0.04 | 36.0 | 0.28 |
| | 14,000 | 0.01 | 100.0 | 0.01 | 14,000 | 0.04 | 39.0 | 0.23 |
| BiLN+RayS a=2 | 4,000 | 0.01 | 99.0 | 0.02 | 4,000 | 0.05 | 32.0 | 0.44 |
| | 8,000 | 0.01 | 100.0 | 0.01 | 8,000 | 0.05 | 35.0 | 0.33 |
| | 14,000 | 0.01 | 100.0 | 0.01 | 14,000 | 0.04 | 36.0 | 0.25 |
| BiLN+RayS b=2 | 4,000 | 0.01 | 100.0 | 0.02 | 4,000 | 0.05 | 33.0 | 0.37 |
| | 8,000 | 0.01 | 100.0 | 0.01 | 8,000 | 0.04 | 37.0 | 0.29 |
| | 14,000 | 0.01 | 100.0 | 0.01 | 14,000 | 0.04 | 39.0 | 0.24 |
| BiLN+RayS b=4 | 4,000 | 0.01 | 93.0 | 0.03 | 4,000 | 0.05 | 32.0 | 0.44 |
| | 8,000 | 0.01 | 100.0 | 0.02 | 8,000 | 0.05 | 35.0 | 0.33 |
| | 14,000 | 0.01 | 100.0 | 0.01 | 14,000 | 0.04 | 38.0 | 0.25 |
| HSJA | 4,000 | 0.01 | 88.0 | 0.06 | 4,000 | 0.09 | 15.0 | 3.57 |
| | 8,000 | 0.01 | 100.0 | 0.02 | 8,000 | 0.08 | 26.0 | 2.41 |
| | 14,000 | 0.01 | 100.0 | 0.01 | 14,000 | 0.06 | 27.0 | 1.69 |
| BiLN+HSJA 16 | 4,000 | 0.02 | 82.0 | 0.03 | 4,000 | 0.07 | 25.0 | 0.68 |
| | 8,000 | 0.01 | 94.0 | 0.01 | 8,000 | 0.06 | 31.0 | 0.47 |
| | 14,000 | 0.01 | 97.0 | 0.01 | 14,000 | 0.05 | 32.0 | 0.37 |
| Sign-OPT | 4,000 | 0.06 | 52.0 | 0.70 | 4,000 | 0.33 | 2.0 | 2.28 |
| | 8,000 | 0.04 | 71.0 | 0.89 | 8,000 | 0.30 | 3.0 | 1.04 |
| | 14,000 | 0.02 | 79.0 | 0.00 | 14,000 | 0.28 | 3.0 | 0.59 |
| BiLN+Sign-OPT 16 | 4,000 | 0.09 | 21.0 | 0.01 | 4,000 | 0.37 | 2.0 | 0.08 |
| | 8,000 | 0.06 | 29.0 | 0.00 | 8,000 | 0.35 | 3.0 | 0.04 |
| | 14,000 | 0.06 | 36.0 | 0.00 | 14,000 | 0.33 | 3.0 | 0.04 |
| AE+Sign-OPT | 4,000 | 0.09 | 20.0 | 0.01 | 4,000 | 0.20 | 8.0 | 0.01 |
| | 8,000 | 0.07 | 31.0 | 0.00 | 8,000 | 0.17 | 11.0 | 0.21 |
| | 14,000 | 0.06 | 35.0 | 0.00 | 14,000 | 0.15 | 11.0 | 0.21 |

Table 1: Comparison at certain query intervals between regular and robust models on CIFAR-10.

| | Imagenet ($\epsilon = 3.0$) | | | | + Smoothing ($\sigma = 0.5, \epsilon \simeq 1.0$) | | | |
|---|---|---|---|---|---|---|---|---|
| | # Queries | Avg. $L_2$ | SR | FID-64 | # Queries | Avg. $L_2$ | SR | FID-64 |
| RayS | 4,000 | 4.11 | 47.0 | 0.22 | 4,000 | 12.30 | 9.6 | 0.77 |
| | 8,000 | 3.17 | 60.0 | 0.14 | 8,000 | 12.23 | 11.5 | 0.77 |
| | 14,000 | 2.55 | 69.0 | 0.09 | 14,000 | 12.19 | 11.5 | 0.77 |
| BiLN+RayS a=2 | 4,000 | 4.55 | 43.0 | 0.32 | 4,000 | 11.95 | 7.8 | 0.61 |
| | 8,000 | 3.54 | 55.0 | 0.19 | 8,000 | 11.86 | 9.1 | 0.61 |
| | 14,000 | 2.77 | 63.0 | 0.11 | 14,000 | 11.81 | 9.1 | 0.61 |
| BiLN+RayS b=2 | 4,000 | 4.07 | 47.0 | 0.22 | 4,000 | 10.27 | 9.0 | 0.39 |
| | 8,000 | 3.15 | 59.0 | 0.13 | 8,000 | 10.20 | 10.0 | 0.39 |
| | 14,000 | 2.52 | 67.0 | 0.08 | 14,000 | 10.14 | 12.0 | 0.39 |
| BiLN+RayS b=4 | 4,000 | 4.51 | 44.0 | 0.24 | 4,000 | 9.46 | 13.4 | 0.39 |
| | 8,000 | 3.50 | 53.0 | 0.16 | 8,000 | 9.41 | 13.4 | 0.38 |
| | 14,000 | 2.84 | 63.0 | 0.10 | 14,000 | 9.38 | 14.6 | 0.39 |
| HSJA | 4,000 | 6.27 | 43.0 | 0.86 | 4,000 | 75.75 | 0.0 | 132 |
| | 8,000 | 3.08 | 62.0 | 0.17 | 8,000 | 74.65 | 0.0 | 130 |
| | 14,000 | 1.90 | 82.0 | 0.04 | 14,000 | 72.92 | 2.1 | 125 |
| BiLN+HSJA 32 | 4,000 | 2.05 | 78.0 | 0.01 | 4,000 | 15.73 | 0.0 | 1.33 |
| | 8,000 | 1.28 | 92.0 | 0.00 | 8,000 | 13.56 | 0.0 | 1.06 |
| | 14,000 | 1.01 | 98.0 | 0.00 | 14,000 | 11.78 | 7.1 | 0.76 |
| Sign-OPT | 4,000 | 28.57 | 26.0 | 0.59 | 4,000 | 71.69 | 4.0 | 147 |
| | 8,000 | 27.04 | 35.0 | 0.36 | 8,000 | 71.45 | 4.0 | 145 |
| | 14,000 | 24.97 | 43.0 | 0.91 | 14,000 | 71.37 | 4.0 | 144 |
| BiLN+Sign-OPT 32 | 4,000 | 5.48 | 44.0 | 0.12 | 4,000 | 23.61 | 6.0 | 2.95 |
| | 8,000 | 3.38 | 74.0 | 0.02 | 8,000 | 23.45 | 6.0 | 2.88 |
| | 14,000 | 2.41 | 81.0 | 0.01 | 14,000 | 23.40 | 6.0 | 2.86 |
| AE+Sign-OPT | 4,000 | 34.82 | 3.0 | 1.73 | 4,000 | 48.97 | 5.0 | 10.20 |
| | 8,000 | 28.01 | 4.0 | 1.16 | 8,000 | 48.89 | 5.0 | 9.92 |
| | 14,000 | 23.77 | 6.0 | 1.16 | 14,000 | 48.89 | 5.0 | 9.91 |

Table 2: Comparison at certain query intervals between regular and robust models on ImageNet.

