# OpenReview forum: "Hard-label Manifolds: Unexpected advantages of query efficiency for finding on-manifold adversarial examples"
_ICLR.cc/2021/Conference — Reject_

### Official Review · AnonReviewer1 · 2020-10-26
**Interesting findings but are they impactful?**

**Rating:** 4
**Confidence:** 2

**Review:**

Summary
-------------
The paper investigates methods that perform hard-label adversarial attacks at the zeroth-order and analyze them from the perspective of generating on-manifold adversarial examples, down-scaling of the input space, and query-efficiency on two datasets, cifar and imagenet.


Pros
------
- The paper is well written.
- The paper tries to address a very important topic of adversarial attacks.
- The paper provides some interesting insights on dimension-reduced attacks, query efficiency and reduced resolution for current methods that use zeroth-order hard-label attacks.

Cons
-------
- Although the insights are interesting, it  is not clear how they can be used to design a new SOTA algorithm that outperforms OPT-Attack,  Sign-OPT, and RayS in terms of identifying an adversarial example with smaller distortion with less queries. This seems to be  a big weakness of this work as the experiments comparing the existing methods are not comprehensive enough to help advance the field of designing better adversarial attacks and ways to counter them.

- It is not clear the claim that down-scaling techniques would lead to worse (farther-away) adversarial examples.

- It is not clear why FID is a good measure for determining whether an adversarial example is an on-manifold example.

- No success rates vs query plots like in [1] which is common in the  literature.

- Code is not provided to verify the results

- It doesn't seem like there were multiple runs of the experiments. It is important to have multiple runs and report the mean and std to get more reliable results.

- Lacks method. The paper does not propose a new method. Instead it compares between few existing methods and makes non-impactful insights on their behavior.

Conclusion
---------------
Although the paper is well written and the insights are interesting, it doesn't seem they are impactful enough to be publishable in ICLR.

However, if the authors convince me otherwise about the impact of these insights in the rebuttal period  I will be happy to increase the score.

Post-rebuttal
------------------

I appreciate the response and the manuscript update, but some of the comments have not been addressed such as the code which has not been provided, and the fact that the results are still very inconsistent like R2 mentioned although the arguments made in this work seem clearer to me. But it is still not clear to me how the results can be used to improve current methods for identifying adversarial examples with smaller distortion and less queries. Thus, I will retain my score as is.

[1] SIGN-OPT: A QUERY-EFFICIENT HARD-LABEL ADVERSARIAL ATTACK. ICLR2020

---

> ### Author Response · Authors · 2020-11-19
> **Response to issues outlined by Reviewer 1**
>
> "Although the insights are interesting, it is not clear how they can be used to design a new SOTA algorithm that outperforms OPT-Attack, Sign-OPT, and RayS in terms of identifying an adversarial example with smaller distortion with less queries. "
>
> As discussed in the general response, rather than proposing a new method, the goal of the work is to provide insight into the manifold traversal behavior of the query-efficient hard-label attacks. There is currently no comprehensive analysis of hard-label attack behavior. Since we show that hard-label attacks find on-manifold examples due to the reduced search fidelity, it is possible to propose an extension of first-order adversarial training schemes for the hard-label setting. This can be based on the noisy manifold distance oracle proposed in Section 5, since the noisy signal differs from the direct signal of the gradient-level setting.
>
>
> "It is not clear the claim that down-scaling techniques would lead to worse (farther-away) adversarial examples."
>
> We attempt to address this with new text (highlighted in red) in Section 3.4. To summarize, the search direction is used directly in the adversarial sample update. If the search fidelity is reduced, the resolution of the noise applied to the sample is consequently reduced, which permits less fine-grained updates.
>
>
> "It is not clear why FID is a good measure for determining whether an adversarial example is an on-manifold example."
>
> In general the real manifold of the datasets is difficult to describe. As we point out in the general response, designers of GANs must rely on metrics that can associate the synthetic output of a GAN with the known statistics of the dataset. FID was shown to offer a stable measure of both low-level and high-level distortion. These distortions correspond to modifications which cause a sample to leave the data manifold. Thus we treat the adversarial samples as synthetically generated samples, which can be compared (using FID) to known data (i.e., the original samples).
>
>
> "No success rates vs query plots like in [1] which is common in the literature."
>
> We have provided the success rate vs. query plots in the Appendix (Section A.2) of the latest rebuttal revision.
>
>
> "Code is not provided to verify the results"
>
> We are in the process of providing an anonymized archive of the codebase before the discussion period is over. We have full intention of publicly releasing the code after the work is published.
>
>
> "It doesn't seem like there were multiple runs of the experiments."
>
> We use a similar experimental procedure as previous hard-label attack papers. The experiments are run over 100 adversarial samples and averaged. In the latest rebuttal revision, we also provide the standard deviation. Thus the plots show the mean and standard deviation over 100 samples. This has been clarified in the second paragraph of Setup in Section 4.1.

---

> ### Author Response · Authors · 2020-11-24
> **Looking forward to discussion**
>
> Dear Reviewer 1,
>
> With Discussion stage 2 closing soon, we are still looking forward to your reply. We are eager to know if our response has addressed the initial concerns about the paper’s impact or SR vs. query plots. We would be glad to discuss any follow-up questions, and appreciate your feedback.

---

### Official Review · AnonReviewer3 · 2020-10-26
**Incomplete work with unclear motivation and poor writing**

**Rating:** 3
**Confidence:** 4

**Review:**

#### Major Issues:
1. The biggest problem is, the author claimed "super-pixels help eliminate the search space of off-manifold adversarial
examples, leading to examples which are truly generalization errors" in the introduction, but never provide convincing proofs for this argument. The analysis in Section 3 is basically based on intuition and is full of holes.
2. It is very obvious that the authors do not have enough time to polish this paper. This problem is even more obvious in the later parts of this work. To name a few: (a) in the contribution part of Section1, the sub-title is missing for the third part. (b) in Section 3.3, the author first term bilinear transformation as BiLIN but refer it as BiLN in later parts. (c) Some notations are absent or inconsistend. For example, the $u_i$ vector in Equation 2 is first denoted as $u_j$. BTW, what does $\beta$ represents? (d) The information provided by the images is very unclear. What does the column of "CIFAR10/ImageNet" represents? Standard training? This makes the reviewer can not catch the main idea of the work. Even after reading this paper, I still do not understand what method has the author proposed as they claimed in their third contribution.
3. The author claimed in the third contribution of their work that their method can encourage on-manifold examples. How will this help improving robustness? Will this decrease the generalization errors?

#### Minor Issues:
1. In Sec 3.4, the author wrote: "Thus, we are interested in the tradeoff between query
efficiency and search fidelity. This introduces our central research question: How does searching over
reduced-dimension increase efficiency, if the search resolution is decreased as a side-effect?" What is the answer to this question. BTW, the author never mention this "trade-off" part in the introduction, which seems very abrupt to me.
2. The Q&A in Sec 4.1 are redundant. It basically rephrase the content in the three contribution.
3. Why select FID-64? How about features in other depth?

---

> ### Author Response · Authors · 2020-11-19
> **Response to issues outlined by Reviewer 3**
>
> "The analysis in Section 3 is basically based on intuition and is full of holes."
>
> We have improved the language of Sections 3 and 4 to offer a concrete explanation of our results. As discussed in the general response, the new text is highlighted in red, green, or blue based on the reviewer it addresses.
>
>
> "It is very obvious that the authors do not have enough time to polish this paper. This problem is even more obvious in the later parts of this work."
>
> We address each of (a), (b), and (c) in the latest rebuttal revision, and have highlighted the fixed text in blue to make the changes clear. With regards to (d), the plot titles for CIFAR-10 and ImageNet have been updated to make the meaning of each column obvious. These plots and associated text have also been updated based on helpful feedback from the other reviewers.
>
>
> "The author claimed in the third contribution of their work that their method can encourage on-manifold examples. How will this help improving robustness? Will this decrease the generalization errors?"
>
> Our empirical results show that the query-efficient hard-label attacks can find on-manifold examples which are not encountered during training. This occurs due to the query-efficient search acting as indirect manifold feedback from the model. The effect is noticeable when attacking robust models, which are already known to leak manifold information in the gradient-level setting given the results of Engstrom et al. [1]. We also argue that the on-manifold samples are not seen during adversarial training of these robust techniques, as the hard-label adversary has a noisy signal from the model (due to the hard-label assumption). Future work could leverage this in a hard-label adversarial training scheme, as an extension of the existing first-order methods to find samples that improve generalization.
>
>
> "Why select FID-64? How about features in other depth?"
>
> We provide an explanation in the Setup of the revision version. To summarize, FID must be calculated full-rank and requires _at least_ 2,048 samples using the original 2048-dimensional feature layer. It is time-prohibitive to generate 2,048 adversarial samples on certain defense schemes (in particlar, certified smoothing ImageNet) due to high time complexity of the defense. To make the analysis consistent, we use the 64-dimensional layer.
>
> [1] S. Santurkar, D. Tsipras, B. Tran, A. Ilyas, L. Engstrom, and A. Madry, “Image Synthesis with a Single (Robust) Classifier,” arXiv:1906.09453 [cs, stat], Aug. 2019.

---

> ### Author Response · Authors · 2020-11-24
> **Looking forward to discussion**
>
> Dear Reviewer 3,
>
> Based on our response and latest rebuttal revision, we are eager to know if there are still concerns about the paper’s analysis. We have tried to address each issue, and are glad to discuss follow-up questions before the Discussion stage 2 is over.

---

### Official Review · AnonReviewer2 · 2020-10-28
**Paper with interesting empirical observations but lacking clear, precise explanations and insights**

**Rating:** 5
**Confidence:** 3

**Review:**

This paper studies hard label adversarial attacks, to specifically examine whether such attacks find adversarial examples that are close to the data manifold. The paper argues this is counter intuitive to our current understanding that such examples leave the data manifold. The paper studies such examples in both regular and robust models on CIFAR and ImageNet datasets. I enjoyed reading the first half of the paper, but the second half and observations from the experiments were quite hard to follow because they are often vague and rely (heavily) on results from recently published work. This makes it challenging to interpret the results clearly. The plots themselves are also quite hard to read in most cases.

Strengths:
+ Understanding the manifold traversal for zeroth order attacks is pertinent. The paper reveals an unexpected benefit for such methods, indicating that these adversarial examples may lie on the manifold, indicating that previous notions that rely explicitly on manifold-projection based strategies can fail defend against such attacks.
+ The paper presents an interesting case study of known attacks on popular benchmark recognition tasks. The central questions in the paper are interesting.
+ The results show to some degree that the variants considered in the paper (proposed by earlier papers) are closer to the manifold (under the FID measure). This improvement is exaggerated when the models are robust, which is very interesting.


Weaknesses:
* While the paper is well written overall, and easy to read, I found the language used to be imprecise which made some parts of it harder to appreciate. The paper studies an important problem, and demonstrates empirically, the benefit of using a query efficient zeroth order hard label attack, yet it seems the paper does not explain why the result is so? The observation is repeatedly referred to as unexpected, and considering it is one of the central contributions of the paper, I think it would benefit if the paper discussed the main insight into why such an observation occurred in the first place, and more importantly what it tells us about adversarial attacks.
* Too many details are left out of the paper, which make the exact contributions of this work harder to understand. For example, there is only a brief mention of the mapping functions like an AE or a BiLIN or how they are used in this context. How is the BiLIN used here? The reduced dimension ZO search is already shown to be more query efficient in Tu et al (2019). Why do BiLIN variants exploit reduced resolution better?
* Is the finding Q3 of this paper or from Tu et al. (2019)? What new insight does the present manuscript bring that Tu et al (2019) do not have?
* Why is FID the right manifold distance measure? It would greatly help if the observation that query efficient ZO attacks find on-manifold samples, was validated independently outside the FID measure. In other words, does this attack render “manifold projection” based defenses useless? If the sample is already on or close to the manifold, in theory these attacks would be exactly reproduced under a manifold-projection defense?
* The explanations of CIFAR-10 results could be expanded further, it is not clear why AE+Sign-OPT does worse. What does the similarities of the FID curves imply? In general, I think it would help if the paper made more explicit what were to be inferred from these plots. They are also hard to read (perhaps a log axis would help?)
* What is the success rate for these attacks? This is only provided in the supplement, but the experiments section refers to them. It would help if they were in the main text as well.
* Some editorial comments: I would recommend taking some of the explanations for BiLIN outperforming AE variants to motivate its use earlier when it is introduced itself.
* in Section 5 “the manifold can be leaked by the loss landscape” what does this mean exactly?



----------- Update after revision ---------------------

Dear Authors,

Thank you for the revised manuscript. The expanded sections have helped me understand the paper more deeply, and place the contributions in context to prior work better than before. I especially appreciate the new experiments and explanations on the Robust Manifold Defense.

A few comments on the revised paper:

* First, the success rate plots which were missing are quite revealing: While the biggest benefit of the reduced resolution query search appears to have been in robust models (Madry et al), the benefit is not reflected as clear in terms of the success rates (which is arguably more critical). For example, in CIFAR-10 experiment in Fig 2(a) while in the Natural case, BiLN+HSJA is slightly more query efficient it appears less successful than plain HSJA in Fig 6(a). (Once again, the plots and figures are extremely illegible).
* For "Robust" CIFAR-10 models and "Natural" ImageNet , there appears to be a considerable gain in both $L_\infty $ distortion *and* success-rate.  For the "Robust" ImageNet case, while there are *huge* gains in terms of $L_\infty $ distortion, there is essentially no discernible difference in the success rate between reduced-resolution and the original attacks.
* by the way there appears to be a typo in Fig 6(a): both BiLN+Sign-OPT and AE+SignOPT are the same color in the plot
* In the new Figure 4 with MNIST experiments, in all the experiments, the base attack appears to be more query efficient than the resolution reduced attack. For a few it appears that before 5000 queries the reduced-resolution are better, but claiming this supports the earlier results isn't as convincing.
* The observations from these plots are confusing:
>"The hard-label attacks consistently outperform against the Robust Manifold defense":
Again not clear how it is claimed that the attack is outperforming the defense without mentioning success rates for this experiment. This also does not clearly outline why the reduced-resolution variants to not match up to the base attacks?
* The reconstruction error plots are also very hard to see -- once again all the reduced-resolution suffer in these experiments. The explanation given is that:
> "Due to the dynamics of the topological hierarchy discussed before, the dimension-reduction is less effective on MNIST"
This is another example of very heavy terminology that makes it hard to understand what the argument is exactly.

Overall, I think the paper needs more clarity in its experiments. The story is muddled by several inconsistent results, some showing improvements with reduced resolution (the finding is surprising and genuinely interesting.. but unfortunately doesn't seem to be consistent; especially when looking at it with the success rate metric)

I think the authors have a convincing motivation and argument that query-efficient hard label attacks achieved with reduced-resolution can be closer to the manifold, this is clear in Robust CIFAR-10, Natural ImageNet models. The paper does a good job of explaining why these are the case, unfortunately the hypothesis doesn't hold up just as well in the other scenarios, which raises the question as to how these can be used in a more realistic scenario -- and more importantly as one of the other reviewers also pointed out -- what the reader must take out of these results to appreciate query efficient attacks better. I think in its current state the experiments portray a more confusing picture than the arguments in the paper suggest. As a result, I will retain my original score.

---

> ### Author Response · Authors · 2020-11-19
> **Response to each issue outlined by Reviewer 2**
>
> We have provided a rebuttal revision which attempts to address the issues given. With regards to the language of the paper, we have made the paper read more precisely in key areas, highlighted in green in the latest rebuttal revision. In particular, Sections 4.1 and 4.2 now contain more direct conclusions about the cause of this behavior. To summarize, the reduced search direction allows discovery of samples which were not encountered in the training data. This is because hard-label attacks rely on a noisier signal from the model in order to generate samples.  The samples can likewise differ from those generated by gradient-level schemes.
>
>
> "How is the BiLIN used here?"
>
> In the latest rebuttal revision, we have updated Section 3.3 to explicitly describe the use of AE and BiLN. Significant changes are shown in green and blue text to address Reviewers 2 and 3 concerns, respectively.
>
>
> "The reduced dimension ZO search is already shown to be more query efficient in Tu et al (2019). Why do BiLIN variants exploit reduced resolution better?"
>
> Tu et al. show (for the score-level setting) an improvement in query efficiency when performing dimension reduction. However, the largest gain comes from dynamically changing the number of search directions $q$ using their proposed algorithm. We leverage dimension reduction alone to control the query efficiency of each attack, since our main interest is measuring the effect of query efficiency in a controlled manner. As a side effect, we determine if the results of Tu et al. extend to the hard-label setting. This is included in Section 4 of the rebuttal revision.
> BiLN variants can search closer to the original samples, since they do not depend on the adversary's description of the manifold (which can be crude in practice using an AE).
>
>
> "Is the finding Q3 of this paper or from Tu et al. (2019)? What new insight does the present manuscript bring that Tu et al (2019) do not have?"
>
> Tu et al. only experiment in the score-level setting. With regards to the effect of dimension reduction on query efficiency alone, we show it can have a large performance gain in the hard-label setting. However this is not our primary contribution, our focus is on the effect of query-efficiency on manifold traversal during the attacks.
>
>
> "Why is FID the right manifold distance measure?"
>
> As discussed in our general response, the real manifold for practical datasets is difficult to describe. Thus designers of GANs are interested in metrics, such as FID, which can measure the quality of synthetic images with respect to known statistics of the real dataset. In this way it can be ensured that the GAN will produce an on-manifold image without relying on subjective human scoring. Similarly we are treating the adversarial samples as synthetically generated images. Since the Inception-V3 model used in FID is never targeted during our attacks, we can ensure that FID does not rely on any internal aspects of the victim models.
>
>
> "It would greatly help if the observation that query efficient ZO attacks find on-manifold samples, was validated independently outside the FID measure. [...]"
>
> Based on this suggestion, we have included an ablation study in Section A.1 of the Appendix on MNIST. We measure the attack effects on a defense that used manifold-projection during adversarial training, and two other defense baselines. A detailed summary can be found in the latest rebuttal revision. We show that under query-efficient hard-label attacks, the manifold-projection defense has a consistently lower adversarial accuracy, despite this defense outperforming the other two baselines in the gradient-level setting (Figure 4 of latest copy). We also measure the pairwise reconstruction distance when using the defender’s autoencoder to project to the manifold, and show that the query-efficient attacks are always closer to a PGD baseline. Notably the PGD baseline has the cleanest signal of manifold information from the model, since it is a gradient-level attack [1].
>
> With regards to explanation of plots, we have labeled the inset plots and improved the exposition of these results in general. These changes are highlighted in the latest rebuttal revision according to the color key provided in the general response.
>
>
> "In Section 5 'the manifold can be leaked by the loss landscape' what does this mean exactly?"
>
> It is shown by Engstrom et al. that in the gradient-level setting, attacks receive better manifold information as the model becomes more well-regularized from adversarial training [1]. We argue that this is the case in the hard-label setting, regardless of model-regularization, as a side-effect of the query-efficient attacks having a lower search fidelity. The lower fidelity makes it more likely to find high-level semantic features.
>
>
> [1] S. Santurkar, D. Tsipras, B. Tran, A. Ilyas, L. Engstrom, and A. Madry, “Image Synthesis with a Single (Robust) Classifier,” arXiv:1906.09453 [cs, stat], Aug. 2019.

---

> ### Author Response · Authors · 2020-11-24
> **Looking forward to discussion**
>
> Dear Reviewer 2,
>
> We are looking forward to any follow-up questions you might have based on our earlier responses. Before Discussion stage 2 ends, we would be glad to address any remaining concerns about the ablation study or the use of FID as a metric in our paper.

---

### Author Response · Authors · 2020-11-16
**Addressing major issues, and changes in first rebuttal revision**

We thank the reviewers for their helpful comments. Reviewers 1 and 2 have pointed out that although the work seems interesting, there is difficulty in understanding the exact cause for the behavior. We have modified the text significantly in order to provide clear explanations. Based on the feedback from Reviewer 2, we have conducted additional experiments to make a concrete connection between the previously proposed Robust Manifold defense by Jalal et al. [1] and our observations in the hard-label setting. Specifically we provide discussion in context of their findings.

We show in the new Figure 4 that, in general, query-efficient hard-label attacks outperform against the Robust Manifold defense, compared to the robust model baselines (Madry et al. and TRADES, which do not leverage manifold projection). This is despite the previous results by Jalal et al. which show that Robust Manifold defends better than those baselines in the gradient-level setting.

The Robust Manifold defense is only shown robust by Jalal et al. on the MNIST and Celeb-A classification task. We show results on MNIST, but FID has not been investigated with MNIST. Thus we rely on the autoencoder checkpoint released by Jalal et al. to measure pairwise distance between the manifold-projected original and adversarial samples. These results are provided in the appendix in Figure 5, which also compare against a PGD attack as baseline. We have provided a comprehensive discussion of these figures to relate the results back to the main text. To summarize, Sign-OPT and HSJA both have smaller pair-wise distance than the PGD attack on the natural model. This supports our main result, which is that query-efficient attacks leverage manifold information, regardless of the model regularization. On the robust models, PGD acts as a best case for manifold information during the attack due to results by Engstrom et al. [4]. This is supported by the results in the new Figure 5, and shows that the best-performing attacks from Figure 4 were closer to the PGD samples in the manifold projection. On the Robust Manifold defense model, we see that hard-label attack samples all lie close to the PGD samples (compared to the baselines in middle columns). The exception is RayS, which behaves differently due to not relying on an explicit gradient estimate.

Another concern is the use of FID as a measure of manifold distance. In general the real manifold is difficult to describe. When GANs generate artificial samples, a measure is needed to measure the quality of the synthetic images. This was the original motivation for the proposal of Inception Distance and later FID. It is shown in [3] that FID can capture high-level distortions which would reduce the quality of synthetic images. Similarly, we are treating the adversarial samples as synthetically generated images, which can be measured using quality metrics such as FID.

We have updated the figures to make them clearer. Inset plots have been labeled and are now referenced in the text with concise explanation of their meaning. We have also updated the titles of each column as pointed out by Reviewer 3 to make it clearer what they represent. The FID score y-axis for robust models is now independently scaled for clarity. In some sections, we have made significant changes to the text in order to make our contributions clearer. These changes are highlighted using the color key found below. The success rate vs. queries are now given in Section A.2 of the Appendix, as suggested by Reviewer 1.

We clarify that we are not necessarily proposing a new method, the goal of the work is provide a new insight into the behavior of existing hard-label attack methods, which is supported by our experiments on CIFAR-10, ImageNet, and MNIST. We follow a similar approach to previous works which have revealed insights rather than new methods [2].

To aid the revision reading, we have color-coded any modified text based on the reviewer it is addressing:
Reviewer 1=red, Reviewer 2=green, Reviewer 3=blue




[1] A. Jalal, A. Ilyas, C. Daskalakis, and A. G. Dimakis, “The Robust Manifold Defense: Adversarial Training using Generative Models,” 2019.

[2] D. Stutz, M. Hein, and B. Schiele, “Disentangling Adversarial Robustness and Generalization,” in IEEE Conference on Computer Vision and Pattern Recognition (CVPR), 2019, p. 12.

[3] M. Heusel, H. Ramsauer, T. Unterthiner, B. Nessler, and S. Hochreiter, “GANs Trained by a Two Time-Scale Update Rule Converge to a Local Nash Equilibrium,” 2018.

[4] S. Santurkar, D. Tsipras, B. Tran, A. Ilyas, L. Engstrom, and A. Madry, “Image Synthesis with a Single (Robust) Classifier,” 2019.

---

### Author Response · Authors · 2020-11-18
**Follow-up for first rebuttal**

To make our work more consistent with the existing literature, we have added the standard deviation to the distortion vs. queries plots, as requested by Reviewer 1. We have added text to the captions of these figures to describe the shaded areas. Regarding the request for a code release, we have full intention to publicly release our code once the work is published. We are currently working on providing the reviewers an anonymized copy of the codebase.

---

### Decision · Program_Chairs · 2021-01-07
**Final Decision**

**Decision:**

Reject

**Comment:**

The paper investigates several properties of adversarial examples obtained by hard-label attacks. There are some interesting findings in this paper, such as the connection between query efficiency and distance to the image manifold. However, all the reviewers think the paper is below the acceptance threshold due to several weaknesses, including insufficient experiments, clarity, and whether the observations made in the paper can benefit query efficiency or quality of hard-label attacks.